

# Probabilistically violating the first law of thermodynamics in a quantum heat engine

**Timo Kerremans[1], Peter Samuelsson[1] and Patrick P. Potts[1,2]\***

**1** Department of Physics and Nanolund, Lund University, Box 118, 221 00 Lund, Sweden.
**2** Department of Physics, University of Basel, Klingelbergstrasse 82, 4056 Basel, Switzerland.

\* patrick.potts@unibas.ch

## Abstract

Fluctuations of thermodynamic observables, such as heat and work, contain relevant information on the underlying physical process. These fluctuations are however not taken into account in the traditional laws of thermodynamics. While the second law is extended to fluctuating systems by the celebrated fluctuation theorems, the first law is generally believed to hold even in the presence of fluctuations. Here we show that in the presence of quantum fluctuations, also the first law of thermodynamics may break down. This happens because quantum mechanics imposes constraints on the knowledge of heat and work. To illustrate our results, we provide a detailed case-study of work and heat fluctuations in a quantum heat engine based on a circuit QED architecture. We find probabilistic violations of the first law and show that they are closely connected to quantum signatures related to negative quasi-probabilities. Our results imply that in the presence of quantum fluctuations, the first law of thermodynamics may not be applicable to individual experimental runs.

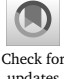

# 1 Introduction

The laws of thermodynamics are cornerstones of modern science, providing fundamental constraints on physically allowed processes [1]. The first law of thermodynamics states that a change in energy can be divided into heat and work

$$\langle \Delta U \rangle = \langle Q \rangle - \langle W \rangle \,, \tag{1}$$

which is a statement of the fundamental law of energy conservation (the signs being chosen such that heat consumed and work produced by a heat engine are positive). The second law of thermodynamics states that entropy never decreases [1,2]

$$\langle \Delta S \rangle \geq 0 \,. \tag{2}$$

In addition, the zeroth law of thermodynamics provides a definition for the concept of temperature, and the third law provides constraints on the behavior at zero temperature.

The theory of thermodynamics as originally developed applies to macroscopic systems, where fluctuations around mean values are irrelevant. For small systems, where fluctuations can no longer be neglected, stochastic thermodynamics [3, 4] provides an extension to the traditional theory. In the presence of fluctuations, entropy production becomes a stochastic

variable described by a probability distribution $P(\Delta S)$ [5]. Importantly, processes with negative entropy production may be observed, *probabilistically* violating the second law of thermodynamics. On the level of these probabilities, a generalized formulation of the second law is provided by fluctuation theorems [3, 6–9]. As a consequence, Eq. (2) still holds, but the left-hand side now denotes the average entropy change. Similar to entropy, in small systems also work and heat fluctuate. In classical systems, energy conservation enforces the first law for every process, that is, there is no probabilistic violation of the first law of thermodynamics.

In quantum systems the situation is less clear. Recently, a considerable effort went into including quantum effects, such as the superposition principle and entanglement, into the theory of thermodynamics [10]. Despite considerable progress, defining the basic thermodynamic observable *work* as a fluctuating quantity still presents a topic of debate [11]. At the heart of this debate stands the active role that the observer takes in quantum theory: while classical fluctuating systems can be described by well-defined trajectories through phase space, the superposition principle prevents such an observer-independent picture. Indeed, it has been shown that any definition of fluctuating work cannot simultaneously fulfill a number of desirable properties that are taken for granted in the classical regime [12]. While work fluctuations are in general affected by an observer, this is not necessarily the case for heat fluctuations. In particular, for weak coupling between system and reservoir, heat is mediated by energy quanta that are exchanged with the reservoir. The number of exchanged quanta is well-defined and independent of any observer. In this case, one may define heat as a fluctuating quantity in complete analogy to classical stochastic systems [13–16].

This illustrates that in a given system, heat and work may behave in a qualitatively different way [17]. Can the first law of thermodynamics then still be expected to hold beyond average values? In this work we show that in quantum systems, the first law of thermodynamics can be violated probabilistically. Such violations may occur when heat and work are independently accessed in a situation where quantum superposition prevents an observer-independent definition for these quantities. Importantly, probabilistic violations of the first law do not imply a violation of energy conservation. Rather, they reflect an uncertainty on our knowledge of energy changes, once they are split into heat and work.

To illustrate these effects, we provide a detailed case study of heat and work fluctuations in the heat engine proposed in Ref. [18], sketched in Fig. 1. There, work may be defined through the time-integral of power. We find probabilistic violations of the first law which are a consequence of the non-commutativity of the power operator with the Hamiltonian. The engine we consider is particularly suitable for our purposes because of multiple reasons: As a thermoelectric device, work fluctuations are directly linked to current fluctuations, a topic that is discussed extensively in the literature [19]. Furthermore, heat and work are carried by photons and electrons respectively, providing a natural separation of these quantities and their fluctuations. Finally, the same architecture was recently used to produce entangled photon beams [20], illustrating the experimental feasibility of the heat engine. We note that the variances of heat and work were investigated in a very similar architecture [21]. As we show below, the probabilistic violations of the first law are only manifested in higher cumulants.

The rest of this article is structured as follows. In Sec. 2, we consider a general setting, introduce definitions for heat and work fluctuations, and discuss the occurence of probabilistic violations of the first law. In the subsequent sections, we turn to a detailed case study to illustrate these violations. In Sec. 3, we introduce the heat engine that is the subject of our case study. Using a local master equation, the laws of thermodynamics follow from a consistent treatment as discussed in Sec. 4. Section 5 illustrates how the fluctuations of heat and work are computed. We present our quantitative results in Sec. 6. Conclusions and an outlook are provided in Sec. 7.

## 2 Probabilistic Violations of the First Law

### 2.1 The Hamiltonian

To set the stage, we first consider a general scenario before providing a detailed case study below. To this end, we consider the Hamiltonian

$$\hat{H}_{\text{tot}}(t) = \hat{H}(t) + \hat{V} + \hat{H}_{\text{B}}, \tag{3}$$

where $\hat{H}(t)$ denotes the Hamiltonian of the system of interest, $\hat{H}_{\text{B}}$ the Hamiltonian of the environment, and $\hat{V}$ the coupling between system and environment. We will be interested in the work produced by the system, the heat provided by the environment, as well as the change in internal energy during the time-interval $[0, \tau]$. To define these quantities and their fluctuations, we consider a way of measuring them. Throughout this manuscript, we consider the weak system-bath coupling regime, where the contribution of $\hat{V}$ to energy changes may be neglected.

### 2.2 Measuring internal energy changes

To measure the changes in internal energy, we consider the two-point measurement scheme [22]. In this scheme, the energy of the system is determined by a projective measurement both at $t = 0$ as well as $t = \tau$. The internal energy change is then determined by

$$\Delta U = E(\tau) - E(0), \tag{4}$$

where $E(t)$ denotes the outcome of the projective energy measurement at time $t$. While often resulting in appealing results (e.g., validity of the Crooks theorem [23]), the two-point measurement has been criticized because the initial energy measurement destroys any initial coherence in the energy basis and is thus not compatible with processes that rely on such initial coherences. Below, we will be interested in a scenario where such initial coherences are unimportant.

### 2.3 Measuring heat

Measuring heat and its fluctuations is experimentally very challenging. A promising route towards measuring heat fluctuations is provided by detecting single energy quanta that are exchanged with the system, which can potentially be achieved by monitoring temperature [24–26]. Very recently, temperature fluctuations, which can be connected to the second cumulant of heat fluctuations, have been observed experimentally [27]. To circumvent the limitations and challenges that come with any specific experimental setup, we consider here again the two-point measurement scheme. To determine the heat exchanged with the environment, its energy is determined by a projective measurement at the beginning and at the end of the time interval $[0, \tau]$ [with outcomes denoted by $E_{\text{B}}(t)$]. The heat provided by the reservoir is then simply given by

$$Q = E_{\text{B}}(0) - E_{\text{B}}(\tau), \tag{5}$$

note the sign difference with respect to Eq. (4).

Below, we are interested in the weak coupling regime and the long-time limit, where any effect due to initial coherence in the system as well as system-bath correlations can be neglected. To ensure the same in the general scenario, we consider an initial state

$$\hat{\rho}_{\text{tot}}(0) = \hat{\rho}_0 \otimes \hat{\rho}_{\text{B}}, \tag{6}$$

with $[\hat{\rho}_0, \hat{H}(0)] = 0$ as well as $[\hat{\rho}_B, \hat{H}_B] = 0$. In this case, both $Q$ and $\Delta U$ can be measured precisely without disturbing the dynamics of the system. As a consequence, heat and internal energy changes take on a well-defined, observer-independent value in each experimental run.

We note that a projective energy measurement of the environment is usually a hopeless task as the environment often comprises a large amount of degrees of freedom. The heat measurement introduced here should thus be understood as a way to define heat, not to measure it in practice. However, as heat can in principle be measured without disturbance (in the weak coupling regime), *any* experimental measurement that avoids disturbance should give the same results within the experimental precision.

## 2.4 Measuring work

To introduce a measurement of work, we consider the quantity of interest being the power output described by the operator $\hat{P} = -\partial_t \hat{H}(t)$, as is the case in thermoelectric devices as the one discussed below. Detailed information on why a two-point measurement scheme for work is not appropriate is given below, in Sec. 2.7. Following Refs. [28–30], we model the measurement by a detector, described by the conjugate variables $\hat{r}$ and $\hat{\pi}$, which is coupled linearly to the power operator during the time interval $[0, \tau]$ by the coupling Hamiltonian

$$\hat{H}_m = s\hat{P}\hat{\pi}, \tag{7}$$

where $s$ denotes the coupling strength. At $t = \tau$, the observable $\hat{r}$ is measured projectively, yielding information about the integrated power and thus the performed work. In practice, the detector could be provided by an LC element [31]. For simplicity, we consider a detector that has no internal dynamics (i.e., no additional term to the Hamiltonian).

This measurement scheme results in the measured distribution [29, 32], see App. A

$$P_m(W) = \int dW' d\gamma \, \mathcal{W}\big([W - W']s, \gamma/s\big) P_\gamma(W') \,, \tag{8}$$

where the subscript m stands for *measurement*, $\mathcal{W}$ denotes the initial Wigner function of the detector, and $P_\gamma(W)$ is a quasi-probability distribution (QPD) that is independent of the detector parameters. This QPD may take on negative values which have been shown to arise from quantum interference effects [33]. The quantity $\gamma$ can be understood as the momentum of the detector (re-scaled by $s$), which acts back on the system through the coupling Hamiltonian given in Eq. (7). The expression for the measured distribution in Eq. (8) can be interpreted as the intrinsic fluctuations of the system, encoded in the QPD, being modified by the detector [34]. This modification presents a trade-off between measurement imprecision and back-action [30]. For a weak measurement (small $s$), $\gamma$ is restricted to small values, implying small back-action. At the same time, the integral over $W'$ describes a convolution with a broad distribution, implying large imprecision. A strong measurement (large $s$) yields small imprecision, as $W - W'$ is restricted to small values. At the same time, a large range for $\gamma$ contributes to the integral implying large back-action. It is interesting to note that while $P_\gamma(W)$ may become negative, back-action and imprecision conspire in a way that always ensures the measured distribution $P_m(W)$ to remain non-negative.

It is instructive to consider an ideal but unphysical detector that has no imprecision and exerts no backaction. Such a detector would be described by a single point in phase space, i.e., $\mathcal{W}(x, p) = \delta(x)\delta(p)$ (as obtained in the classical limit of the ground state of a harmonic oscillator [35]). In that case, the measured distribution reduces to

$$P(W) \equiv P_{\gamma=0}(W), \tag{9}$$

which can be interpreted as describing the work fluctuations in the absence of a measurement. This is the distribution we will use for characterizing work fluctuations.

Even though $P(W)$ cannot describe a measurement by itself (as it may become negative), it can nevertheless be accessed experimentally by considering a weak measurement, where $s \to 0$. In this case, the integrand in Eq. (8) is only finite for vanishingly small $\gamma$ and we may replace $P_\gamma(W)$ by $P(W)$. The measured distribution then simplifies to

$$P_{\mathrm{m}}(W) = \frac{1}{\sqrt{2\pi}\sigma} \int dW' e^{-\frac{(W-W')^2}{2\sigma^2}} P(W'), \tag{10}$$

where we assumed an initial Gaussian distribution for the detector position $\hat{r}$ with variance $s\sigma$. Clearly, if $\sigma$ is known, the distribution $P(W)$ can be recovered from the measurement. This becomes particularly apparent by considering the cumulants (i.e., the mean, variance, skewness, etc.). In the weak measurement limit, the cumulants of the measured distribution are related to the cumulants of $P(W)$ by the simple relation

$$\langle\langle W^k \rangle\rangle_{\mathrm{m}} = \delta_{k,2}\sigma^2 + \langle\langle W^k \rangle\rangle. \tag{11}$$

The only effect the measurement has is an increase of the variance by $\sigma$, which describes the measurement noise. We note that the QPD described here has been used extensively to characterize electronic transport in phase-coherent systems [19, 36, 37].

In the presence of initial coherences, where our definition for internal energy fluctuations becomes problematic, one may use a similar approach as for work fluctuations [32, 38, 39]. Fluctuations in $\Delta U$ are then described by a QPD which reduces to the distribution obtained from a two-point measurement scheme in the absence of initial coherences.

## 2.5 The classical regime

To understand probabilistic violations of the first law, it is instructive to first consider a scenario where they are absent, which we here denote by the *classical regime*. We will be interested in the distribution that describes a joint measurement of heat, work, and internal energy changes, $P_{\mathrm{m}}(Q, W, \Delta U)$. Just as for the work measurement, this distribution can be written as a convolution between a QPD and the Wigner function of the detector used for measuring work (see App. A for details). In the classical regime, we make the assumptions

$$[\hat{H}(t), \hat{H}(t')] = 0, \qquad [\hat{H}(t) + \hat{H}_{\mathrm{B}}, \hat{V}] = 0, \tag{12}$$

for all times $t$ and $t'$. The first assumption ensures that no coherences in the energy-basis are created, such that the system can always be described by the populations of states with well-defined energy. The second assumption ensures that the energy stored in the coupling does not influence the thermodynamics. Alternatively, one could consider a weak coupling approximation, where the thermodynamics to lowest order in the coupling strength is not affected by the energy stored in the coupling.

Under these approximations, we find that the work measurement does not exert any back-action, such that the QPD is independent of $\gamma$. The relation between the measured distribution and the QPD then simplifies to

$$P_{\mathrm{m}}(Q, W, \Delta U) = \int dW' \frac{e^{-\frac{(W-W')^2}{2\sigma^2}}}{\sqrt{2\pi}\sigma} P(Q, W', \Delta U), \tag{13}$$

which holds for all measurement strengths. We furthermore find that the QPD is non-negative and obeys

$$P(Q, W, \Delta U) = P(Q, \Delta U)\delta(W - Q + \Delta U), \tag{14}$$

where $P(Q, \Delta U)$ denotes the distribution obtained from measuring heat and internal energy changes as outlined above, in the absence of any detector for work. Equation (14) implies that in the classical regime, the first law is not only valid on average but holds on the level of the QPD. For the cumulants, this equation implies

$$\langle\langle Q^j W^k \Delta U^l \rangle\rangle = \langle\langle Q^j (Q - \Delta U)^k \Delta U^l \rangle\rangle. \qquad (15)$$

The measured distribution however may not obey an equation analogous to Eq. (14). This is simply a consequence of the fact that the work measurement may be imprecise. Indeed, we find that the measured distribution is proportional to $\delta(W - Q + \Delta U)$ only when $\sigma \to 0$, i.e., in the strong measurement limit where the measured distribution reduces to the QPD (in the classical regime). Fortunately, in case a strong measurement is not available, one may recover the same cumulants from a weak measurement, where we find the relation

$$\langle\langle Q^j W^k \Delta U^l \rangle\rangle_{\mathrm{m}} = \delta_{k,2}\sigma^2 + \langle\langle Q^j W^k \Delta U^l \rangle\rangle. \qquad (16)$$

If the detector parameter $\sigma$ (characterizing measurement noise) is known, one may thus determine the cumulants of the QPD from a weak measurement and verify the validity of the first law using Eq. (15).

## 2.6 Probabilistic violations of the first law

In this work, we are interested in scenarios where the first law does *not* hold on the level of the distribution describing the joint work, heat, and internal energy fluctuations. Due to measurement imprecision and backaction, the measured distribution will in general not respect the first law: Finite imprecision introduces randomness in the measurement outcomes while backaction may introduce energy exchanges between the system and the detector which are not being detected. This implies that we should expect separate measurements of heat, work, and internal energy to result in statistics that do not respect the first law, except in the rare occasions when these quantities can be measured precisely without disturbing the system.

To circumvent these issues, we consider the QPD as the relevant distribution. As mentioned before, this distribution describes the system in the absence of a measurement but can still be recovered by a weak, non-invasive measurement. Furthermore, it obeys the first law in the classical regime. We thus define probabilistic violations of the first law to be present whenever

$$P(Q, W, \Delta U) \neq P(Q, \Delta U)\delta(W - Q + \Delta U). \qquad (17)$$

In terms of the cumulants, this implies that there exist values for $j, k, l$, such that

$$\langle\langle Q^j W^k \Delta U^l \rangle\rangle \neq \langle\langle Q^j (Q - \Delta U)^k \Delta U^l \rangle\rangle. \qquad (18)$$

Such violations can be expected if any of the assumptions in the last subsection are lifted. In particular, if $[\hat{H}(t), \hat{H}(t')] \neq 0$, the fundamental backaction vs imprecision tradeoff cannot be circumvented and the first law is no longer guaranteed on the level of the QPD. Below, we provide a detailed case study of a heat engine where such probabilistic first law violations are present. Alternatively, such violations may occur when the coupling energy between system and reservoir cannot be neglected. In this case, our definitions for heat and internal energy are no longer justified and a strong coupling analysis should be employed [40–42].

An obvious but crucial requirement for observing first-law violations is that heat, work, and internal energy are determined by separate measurements. Otherwise, one may only measure heat and internal energy and simply determine work by assuming the validity of the first law [43–45]. As there is an ongoing debate on how to define work as a fluctuating quantity [11,22,46,47], the choice for the work measurement takes on a crucial role. While we consider

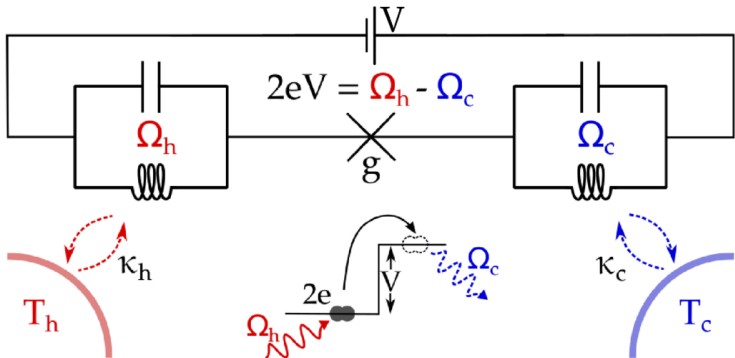

Figure 1: Quantum heat engine based on a voltage biased superconducting circuit. Two single-mode microwave resonators (LC-circuits) with frequencies $\Omega_h$ and $\Omega_c$ are coupled via a Josephson junction, with an effective coupling strength $g$. Each resonator is further coupled, with strength $\kappa_h$ and $\kappa_c$ respectively, to a thermal reservoir kept at temperatures $T_h \geq T_c$. While the inset explains the average behavior of heat and work, it fails to explain the quantum behavior of their fluctuations.

a particular definition for work fluctuations, we believe similar conclusions to hold for other approaches where work, heat, and internal energy are determined by separate measurements. However, when these quantities are defined as conditional averages, conditioned on the same measurements [48–51], the first law holds on the level of probabilities by construction.

## 2.7 Why no two-point measurement scheme for work?

As we are employing the two-point measurement scheme both for heat as well as for internal energy, one may ask why it should not be applied for measuring work as well. To address this question, we note that the two-point measurement scheme for work can be applied in two different ways. The first way is provided by projectively measuring $\hat{H}_{\text{tot}}(t)$ given in Eq. (3) at the beginning and at the end of the time-interval $[0, \tau]$. In the weak coupling regime, where the contribution of the coupling $\hat{V}$ to the energy changes is negligible, this is equivalent to simultaneously performing a two-point measurement scheme on the system and on the bath. This measurement of work is thus equivalent to measuring heat and internal energy simultaneously, implicitly relying on the first law. The first way of implementing a two-point measurement scheme for work does therefore not constitute a separate measurement of work. When considering a heat engine, a direct way of measuring its work output is arguably necessary in order to call the output of the engine useful.

The second way of implementing the two-point measurement scheme is to resort to a time-independent description, by explicitly considering the degrees of freedom that render the Hamiltonian in Eq. (3) time-dependent. These degrees of freedom provide a work-storage device. In the scenario considered below, this is particularly illuminating as the work-storage device is provided by a Josephson junction, where work is carried by electrons that move against a voltage bias, just like when a battery is being charged. One may then envision a two-point measurement scheme applied on the work-storage device. However, to result in a time-dependent Hamiltonian, the work-storage device usually contains a considerable amount of coherence in its initial state. The first projective measurement in the two-point measurement scheme would destroy this coherence, considerably influencing the dynamics of the system and rendering the description in terms of a time-dependent Hamiltonian invalid. For a work-storage device provided by a Josephson junction, this is discussed in more detail at the end of Sec. 5.2.

## 3   The Heat Engine

To investigate the joint fluctuations of heat and work, as well as the occurence of probabilistic violations of the first law, we consider the heat engine proposed in Ref. [18], see Fig. 1. The engine consists of a voltage biased superconducting circuit that defines two single-mode microwave resonators with frequencies $\Omega_h$ and $\Omega_c$. Each resonator is in radiative contact with a thermal reservoir kept at temperatures $T_h \geq T_c$ respectively. A Josephson junction in series mediates a coupling between the resonators by the exchange of photons with tunneling Cooper pairs [52, 53]. Choosing the voltage bias such that (we set $\hbar = 1$)

$$2eV = \Omega_h - \Omega_c \,, \tag{19}$$

allows Cooper pairs to tunnel against the voltage bias by absorbing photons from the resonator with frequency $\Omega_h$ (henceforth denoted as the hot resonator) and emitting photons to the resonator with frequency $\Omega_c$ (the cold resonator). When operating the system as a thermoelectric heat engine, the temperature gradient drives a net flow of photons from the hot to the cold reservoir. This heat flow drives Cooper pairs, i.e., a supercurrent, against the voltage bias, hence producing electrical work. As the supercurrent is dissipationless, it carries no entropy and all the heat is carried by the photons, while the work is provided by the Cooper pairs. This separation of heat and work is very useful when defining these quantities as fluctuating variables.

It is convenient to introduce a simplified pictorial representation of the heat-to-work conversion process (see the inset in Fig. 1): First, a photon enters the hot resonator from the hot reservoir. This photon is then converted into a photon in the cold resonator by a Cooper pair tunneling against the voltage bias. Finally, the photon leaves the system into the cold reservoir. In this process the electrical work performed by the Cooper pair is $2eV$, the heat provided by the hot reservoir is $\Omega_h$, and the heat emitted into the cold reservoir is $\Omega_c$. It is tempting to use this representation to describe the full statistics of heat and work. However, we find it to be an oversimplified picture: While it describes the behavior of mean values well, it fails to capture the behavior of heat and work fluctuations due to the coherent nature of the heat-to-work conversion process.

For a quantitative description, we model the system by the Hamiltonian (for details on the derivation and the involved approximations, see Ref. [18])

$$\hat{H} = \sum_{\alpha=h,c} \Omega_\alpha \hat{a}_\alpha^\dagger \hat{a}_\alpha + g \left[ \hat{a}_c^\dagger \hat{a}_h e^{2ieVt} + \hat{a}_h^\dagger \hat{a}_c e^{-2ieVt} \right] \,, \tag{20}$$

where the time-dependence arises from the dc Josephson effect, which turns a time-independent voltage into a time-dependent phase, see [54] for details. We note that a Hamiltonian of this form has been introduced for a heat engine by Kosloff in 1984 [55]. The couplings of the hot and cold resonators to the reservoirs are characterized by $\kappa_h$ and $\kappa_c$, respectively. Considering throughout the paper the regime

$$g, \kappa_\alpha \ll \Omega_\alpha \,, \tag{21}$$

the dynamics of the system can be described by a local Master equation (without any constraint on the ratio $g/\kappa_\alpha$) [56]

$$\partial_t \hat{\rho} = -i[\hat{H}, \hat{\rho}] + \mathcal{L}_h \hat{\rho} + \mathcal{L}_c \hat{\rho} \,, \tag{22}$$

where

$$\mathcal{L}_\alpha \hat{\rho} = \kappa_\alpha (n_B^\alpha + 1) \mathcal{D}[\hat{a}_\alpha] \hat{\rho} + \kappa_\alpha n_B^\alpha \mathcal{D}[\hat{a}_\alpha^\dagger] \hat{\rho} \,, \tag{23}$$

with $\mathcal{D}[\hat{A}]\hat{\rho} = \hat{A}\hat{\rho}\hat{A}^\dagger - \{\hat{A}^\dagger\hat{A}, \hat{\rho}\}/2$, and the Bose-Einstein distribution

$$n_B^\alpha = \frac{1}{\exp(\Omega_\alpha/k_B T_\alpha) - 1} \ . \tag{24}$$

The first (second) term on the right hand side in Eq. (23) corresponds to the process where one photon is emitted to (absorbed from) the thermal reservoir.

# 4  The laws of Thermodynamics

Local master equations have been criticized for not being thermodynamically consistent, which may result in violations of the second law of thermodynamics [57–59]. These violations have been shown to be small (i.e., of the order of terms that are neglected in deriving a local master equation) [60, 61]. Here we show how the smallness of $g/\Omega_\alpha$, a requirement for the validity of Eq. (22), can be exploited to obtain thermodynamic consistency. To this end, we drop the coupling term in Eq. (20) in the thermodynamic bookkeeping, introducing a *thermodynamic* Hamiltonian

$$\hat{H}_{\text{TD}} = \Omega_h \hat{a}_h^\dagger \hat{a}_h + \Omega_c \hat{a}_c^\dagger \hat{a}_c \ . \tag{25}$$

This Hamiltonian will be used to define the internal energy of the system

$$\langle U \rangle \equiv \text{Tr}\{\hat{H}_{\text{TD}}\hat{\rho}\} . \tag{26}$$

Importantly, $\hat{H}_{\text{TD}}$ is only used for the thermodynamic bookkeeping, the dynamics is still described by Eq. (22) with the full Hamiltonian given in Eq. (20), including a finite value for $g$. We note that this is consistent with a microscopic derivation of the local master equation, which implies that the energy exchanged with the reservoirs cannot be resolved on the scale of $g$ and $\kappa_\alpha$ [56]. We now demonstrate that this approach results in thermodynamic consistency. We note that our treatment here is a specific implementation of a generic approach, presented in Ref. [62]. For a different approach to treating the term dropped in Eq. (25), see Ref. [63].

## 4.1  The zeroth law

The zeroth law of thermodynamics states that if two bodies are in equilibrium with a third, they are also in equilibrium with each other. Here, the three bodies are the two reservoirs and the system. Equilibrium is obtained by setting the temperatures equal $T = T_c = T_h$ and removing any voltage bias $V = 0$. As our model is only justified when Eq. (19) holds, it may only describe equilibrium when $\Omega_h = \Omega_c$. For these equilibrium conditions, the steady-state solution of Eq. (22) is found to be the Gibbs state at temperature $T$ with respect to $\hat{H}_{\text{TD}}$

$$\hat{\rho}_\beta(\hat{H}_{\text{TD}}) = \frac{e^{-\beta\hat{H}_{\text{TD}}}}{\text{Tr}\left\{e^{-\beta\hat{H}_{\text{TD}}}\right\}} \ , \tag{27}$$

where $\beta = 1/k_B T$. We thus find that the zeroth law holds, and that thermal equilibrium is characterized by $\hat{H}_{\text{TD}}$. We note that this is indeed the expected form of the equilibrium state within our approximations

$$\hat{\rho}_\beta(\hat{H}_{\text{TD}}) \overset{g \ll \Omega_\alpha}{\simeq} \hat{\rho}_\beta(\hat{H}) \overset{\kappa_\alpha \ll \Omega_\alpha}{\simeq} \text{Tr}_{\text{B}}\{\hat{\chi}_\beta\}, \tag{28}$$

where $\hat{\chi}_\beta$ denotes the Gibbs state with respect to the total Hamiltonian, including the reservoirs, and $\text{Tr}_{\text{B}}$ denotes the partial trace over the reservoir degrees of freedom [64, 65].

## 4.2 The first law

To formulate the first law of thermodynamics, we require definitions for heat and work. In thermoelectric devices work is usually accessed through the electrical current, which is related to the power operator

$$\hat{P} \equiv -\partial_t \hat{H} = 2eV\hat{I}, \tag{29}$$

where

$$\hat{I} = -ig\left[\hat{a}_c^\dagger \hat{a}_h e^{2ieVt} - \hat{a}_h^\dagger \hat{a}_c e^{-2ieVt}\right], \tag{30}$$

is the particle-current operator for tunneling Cooper pairs. We note that we are only interested in the time-averaged dc-current, and do not consider any ac-current arising from the Josephson effect. The average power is the time derivative of the average work and reads

$$\langle \dot{W} \rangle \equiv \text{Tr}\{\hat{P}\hat{\rho}\} = -i\text{Tr}\{[\hat{H}, \hat{H}_{\text{TD}}]\hat{\rho}\}, \tag{31}$$

where the dot denotes a time-derivative. Next, we define the average heat current from reservoir $\alpha$ to the system as

$$\langle \dot{Q}_\alpha \rangle \equiv \text{Tr}\{\hat{H}_{\text{TD}}\mathcal{L}_\alpha \hat{\rho}\}. \tag{32}$$

We note that we used the thermodynamic Hamiltonian to define heat, in consistency with the discussion above. We return to the consistency of these definitions with Sec. 2 below. Using Eq. (26), it is then straightforward to show that the first law of thermodynamics holds

$$\langle \dot{U} \rangle = \langle \dot{Q}_h \rangle + \langle \dot{Q}_c \rangle - \langle \dot{W} \rangle. \tag{33}$$

## 4.3 The second law

The second law of thermodynamics states that the entropy production cannot be negative. Using Eq. (32) for the average heat flow, we find

$$\dot{S} = -k_B \partial_t \text{Tr}\{\hat{\rho}\ln\hat{\rho}\} - \frac{\langle \dot{Q}_h \rangle}{T_h} - \frac{\langle \dot{Q}_c \rangle}{T_c} = k_B \sum_{\alpha=h,c} \text{Tr}\left\{[\mathcal{L}_\alpha \hat{\rho}]\left(\ln\hat{\rho}_{\beta_\alpha} - \ln\hat{\rho}\right)\right\} \geq 0, \tag{34}$$

where we used Spohn's inequality [66] which relies on $\mathcal{L}_\alpha \hat{\rho}_{\beta_\alpha} = 0$ [where $\hat{\rho}_{\beta_\alpha} = \hat{\rho}_{\beta_\alpha}(\hat{H}_{\text{TD}})$, cf. Eq. (27)]. Once again, the use of $\hat{H}_{\text{TD}}$ is crucial for obtaining a consistent thermodynamic bookkeeping.

We have thus shown that our approach is thermodynamically consistent in the sense that it yields average values for heat, work, and internal energy changes that satisfy the laws of thermodynamics. We stress that these laws make statements about mean values and may not simply be taken over to fluctuating quantities.

We note that the weak system-bath coupling regime has been noted to violate the third law of thermodynamics for time-dependent Hamiltonians [67]. In the limit we consider (where $g \ll \Omega_\alpha$), this effect is expected to be negligible.

## 5 Fluctuations: full counting statistics

We now expand our analysis from average quantities to the full statistics of heat and work. To this end, we consider $Q_\alpha$ as the heat exchanged with reservoir $\alpha$ and $W$ as the work produced in the time interval $[0, \tau]$. We do not consider changes in internal energy, as these become negligible in the long-time limit considered below (see App. C). Interestingly, the heat and work distributions are fully determined by the statistics of transferred photons and Cooper pairs respectively, i.e., the *full counting statistics* of these particles.

## 5.1 Heat fluctuations - counting photons

We first focus on the statistics of heat fluctuations. As mentioned above, the heat flow is exclusively carried by photons exchanged with the environment. To lowest order in $g/\Omega_\alpha$ and $\kappa_\alpha/\Omega_\alpha$, every photon exchanged with reservoir $\alpha$ carries an equal amount of energy, $\Omega_\alpha$. Thus, the heat exchanged with reservoir $\alpha$ is fully determined by the number of photons exchanged with the reservoir, denoted by $q_\alpha$, and we can identify $Q_\alpha = q_\alpha\Omega_\alpha$ such that

$$P(Q_h, Q_c) = \sum_{q_h, q_c} \delta(Q_h - q_h\Omega_h)\delta(Q_c - q_c\Omega_c)P(\boldsymbol{q}), \tag{35}$$

where $\boldsymbol{q} = (q_h, q_c)$. We stress that $q_\alpha$ denotes the *net* number of photons exchanged with reservoir $\alpha$ during the time interval $[0, \tau]$. The sign is chosen such that a positive $q_\alpha$ denotes photons entering the system.

The distribution $P(\boldsymbol{q})$, known as the full counting statistics of the photons, can be obtained by introducing the photon counting fields $\boldsymbol{\chi} = (\chi_h, \chi_c)$ in the master equation [68–70]

$$\partial_t \hat{\rho}(\boldsymbol{\chi}) = -i[\hat{H}, \hat{\rho}(\boldsymbol{\chi})] + \mathcal{L}_h^{\chi_h}\hat{\rho}(\boldsymbol{\chi}) + \mathcal{L}_c^{\chi_c}\hat{\rho}(\boldsymbol{\chi}), \tag{36}$$

with the superoperators

$$\mathcal{L}_\alpha^{\chi_\alpha}\hat{\rho} = \kappa_\alpha(n_B^\alpha + 1)\left[e^{i\chi_\alpha}\hat{a}_\alpha\hat{\rho}\hat{a}_\alpha^\dagger - \frac{1}{2}\left\{\hat{a}_\alpha^\dagger\hat{a}_\alpha, \hat{\rho}\right\}\right] + \kappa_\alpha n_B^\alpha\left[e^{-i\chi_\alpha}\hat{a}_\alpha^\dagger\hat{\rho}\hat{a}_\alpha - \frac{1}{2}\left\{\hat{a}_\alpha\hat{a}_\alpha^\dagger, \hat{\rho}\right\}\right]. \tag{37}$$

The quantity $\hat{\rho}(\boldsymbol{\chi})$ is directly related to the cumulant generating function

$$\mathcal{S}(\boldsymbol{\chi}) = \ln\left[\text{Tr}\{\hat{\rho}(\boldsymbol{\chi})\}\right], \tag{38}$$

which in turn defines the heat distribution

$$P(\boldsymbol{q}) = \int_0^{2\pi}\frac{d\chi_c}{2\pi}\int_0^{2\pi}\frac{d\chi_h}{2\pi}e^{\mathcal{S}(\boldsymbol{\chi}) + i\boldsymbol{\chi}\cdot\boldsymbol{q}}. \tag{39}$$

Note that the $2\pi$ periodicity of $\hat{\rho}(\boldsymbol{\chi})$ reflects the fact that its Fourier transform, $P(\boldsymbol{q})$, is a discrete distribution, taking only finite values for integer values of $q_\alpha$. It is straightforward to show that the average values obtained from this distribution agree with Eq. 32. Furthermore, under Born-Markov approximations, the heat statistics obtained from the full counting statistics can be shown to be the same as the statistics obtained from a two-point measurement scheme applied to each reservoir [14–16].

From the cumulant generating function, the cumulants of the distribution $P(\boldsymbol{q})$ may be derived, with the first, second, and third cumulant providing the mean, variance, and skewness of the distribution. The $k$-th cumulant is obtained by

$$\langle\langle q_\alpha^k\rangle\rangle = i^k\partial_{\chi_\alpha}^k\mathcal{S}(\boldsymbol{\chi})|_{\boldsymbol{\chi}=0}. \tag{40}$$

In the long-time limit, $\mathcal{S}(\boldsymbol{\chi})$, and thus all cumulants, become linear in time, see App. C. In this limit, any entropy change in the system becomes negligible and the second law given in Eq. (34) puts a restriction on the average values

$$\frac{\langle\dot{Q}_h\rangle}{T_h} + \frac{\langle\dot{Q}_c\rangle}{T_c} \leq 0 \iff \langle\langle q_h\rangle\rangle\frac{\Omega_h}{T_h} + \langle\langle q_c\rangle\rangle\frac{\Omega_c}{T_c} \leq 0. \tag{41}$$

It is well known that this restriction only holds for the mean values and does not carry over to fluctuating quantities, i.e.,

$$P(\boldsymbol{q}) \neq P(\boldsymbol{q})\theta\left(-q_h\beta_h\Omega_h - q_c\beta_c\Omega_c\right), \tag{42}$$

where $\theta(x)$ denotes the Heaviside theta function that is equal to one for $x \geq 0$ and zero otherwise. Indeed, for fluctuating systems, the second law is generalized by the fluctuation theorem [8], which in the long-time limit reads (the validity of this equality for our system is shown below)

$$\frac{P(-\boldsymbol{q})}{P(\boldsymbol{q})} = e^{q_h \beta_h \Omega_h + q_c \beta_c \Omega_c} \,. \tag{43}$$

Using Jensen's inequality, the second law, as stated in Eq. (41), may be recovered from the fluctuation theorem.

Finally, we note that in the long-time limit, we find (see App. C)

$$P(\boldsymbol{q}) \propto \delta_{q_h, -q_c} \,. \tag{44}$$

Because there is no photon accumulation (or generation) within the system, the statistics of heat is fully determined by the photons which traverse the system and it is sufficient to consider a single counting field for heat.

## 5.2 Work fluctuations - counting electrons

In our system work is exclusively performed by the supercurrent carried by Cooper pairs tunneling across the Josephson junction. We denote the net number of Cooper pairs that tunneled against the voltage bias in the time-interval $[0, \tau]$ by $w$. The work provided in this time-interval may then be written as $W = 2eVw$ implying

$$P(W) = \sum_{w} \delta(W - 2eVw)P(w) \,, \tag{45}$$

where $P(W)$ denotes the QPD discussed in Sec. 2.4. For phase-coherent systems, the full counting statistics is obtained by introducing a counting field $\lambda$ in the master equation as [29,36] (see Ref. [32] for the connection to the measurement scheme outlined in Sec. 2.4)

$$\partial_t \hat{\rho}(\lambda) = -i[\hat{H}, \hat{\rho}(\lambda)] - \frac{i\lambda}{2}\{\hat{I}, \hat{\rho}(\lambda)\} + \sum_{\alpha=c,h} \mathcal{L}_\alpha \hat{\rho}(\lambda) \,, \tag{46}$$

where the current operator $\hat{I}$ is defined in Eq. (30). In analogy to Eqs. (38) and (39), we introduce a cumulant generating function

$$\mathcal{S}(\lambda) = \ln\left[\text{Tr}\{\hat{\rho}(\lambda)\}\right] \,, \tag{47}$$

which defines the work distribution

$$P(w) = \int_{-\infty}^{\infty} \frac{d\lambda}{2\pi} e^{\mathcal{S}(\lambda) + i\lambda w} \,. \tag{48}$$

It is again straightforward to show that this distribution provides the average value given in Eq. (31). The cumulants of $P(w)$ can then be obtained in analogy to Eq. (40)

$$\langle\langle w^k \rangle\rangle = i^k \partial_\lambda^k \mathcal{S}(\lambda)\big|_{\lambda=0} \,. \tag{49}$$

In contrast to the photon distribution, $P(w)$ is a continuous function. This is related to the fact that the Cooper pairs cannot be counted one by one without disturbing the dynamics. To appreciate this, one may consider a two-point measurement scheme for counting Cooper pairs, in analogy to the discussion on measuring heat in Sec. 2.3. In this approach, the total number of electrons on one side of the Josephson junction is measured at the initial and final times. Here we consider a Josephson junction that has a well defined phase [71,72]. As the

phase is conjugate to the particle number, a measurement of the number of electrons disturbs the phase and would strongly affect the dynamics of the heat engine. We note that a different approach for counting electrons has been employed in a similar system [73]. As shown in Sec. 2.4, the approach taken here is motivated by its connection to an explicit measurement of power.

## 5.3 Joint heat and work fluctuations

We now turn to the distribution $P(\boldsymbol{q}, w)$, providing a joint description of heat and work fluctuations. Because heat is carried only by photons and work only by Cooper pairs, this distribution may be calculated by including counting fields for both photons and Cooper pairs

$$\partial_t \hat{\rho} = -i[\hat{H}, \hat{\rho}] - \frac{i\lambda}{2}\{\hat{I}, \hat{\rho}\} + \mathcal{L}_h^{\chi_h}\hat{\rho} + \mathcal{L}_c^{\chi_c}\hat{\rho}, \tag{50}$$

where we dropped the counting field dependence of $\hat{\rho}$ for ease of notation. The cumulant generating function is then introduced in the usual way

$$\mathcal{S}(\boldsymbol{\chi}, \lambda) = \ln[\mathrm{Tr}\{\hat{\rho}(\boldsymbol{\chi}, \lambda)\}]. \tag{51}$$

Setting $\lambda = 0$ ($\boldsymbol{\chi} = 0$), we recover the cumulant generating function for photons (Cooper pairs) respectively. The joint distribution can then be written as

$$P(\boldsymbol{q}, w) = \int_0^{2\pi} \frac{d\chi_c}{2\pi} \frac{d\chi_h}{2\pi} \int_{-\infty}^{\infty} \frac{d\lambda}{2\pi} e^{\mathcal{S}(\boldsymbol{\chi}, \lambda) + i\lambda w + i\boldsymbol{\chi}\cdot\boldsymbol{q}}, \tag{52}$$

and the cumulants may be obtained by

$$\langle\langle q_h^k q_c^l w^m \rangle\rangle = i^{k+l+m} \partial_{\chi_h}^k \partial_{\chi_c}^l \partial_\lambda^m \mathcal{S}(\boldsymbol{\chi}, \lambda) \big|_{\boldsymbol{\chi}=0, \lambda=0}. \tag{53}$$

Throughout, we will be interested in the long-time limit, where the cumulants grow linearly in time and where $q_h = -q_c$. The fluctuations in the internal energy may then safely be neglected (cf. App. C) and $P(q, w)$ provides a complete description of the fluctuating energy flows through the system, where $q = q_h$ denotes the number of photons that went from the hot to the cold reservoir. In this case, the first law constrains the mean values to satisfy

$$\langle \dot{W} \rangle = \langle \dot{Q} \rangle \iff \langle\langle w \rangle\rangle = \langle\langle q \rangle\rangle, \tag{54}$$

where $Q = Q_h + Q_c = (\Omega_h - \Omega_c)q$. It is tempting to take over this statement of energy conservation to fluctuating quantities, assuming that the work fluctuations can be completely described by the heat fluctuations. However, this is not generally possible in quantum systems and we find probabilistic violations of the first law, i.e.,

$$P(w, q) \neq P(q)\delta(w - q). \tag{55}$$

This condition may also be expressed in terms of the cumulants

$$\exists \ k \ \text{s.t.} \ \langle\langle (w - q)^k \rangle\rangle \neq 0. \tag{56}$$

We stress that the internal energy fluctuations cannot restore the first law because the corresponding cumulants scale differently with time, see App. C.

Investigating a particular heat engine allows for a better understanding of the origin of these violations. In the present system, work is probed via the power (or electrical current) operator, as is usually the case in thermoelectric devices. Because the power operator does not commute with the Hamiltonian $[\hat{H}, \hat{P}] \neq 0$, the Heisenberg uncertainty principle implies

that any information acquired on the power restricts our knowledge on energy. The probabilistic violations of the first law reflects this fundamental limitation in the joint knowledge of work and energy and should not be seen as a violation of energy conservation. As no such limitation applies for classical systems, probabilistic first law violations are a purely quantum phenomenon. In contrast to fluctuations, mean values can quite generally be obtained without disturbing the dynamics, ensuring that the first law is still respected on average.

# 6  Results

In order to explicitly calculate the heat and work distributions, we follow the work of Clerk and Utami [74] and cast the master equation including counting fields [cf. Eq. (50)] into an equation of motion for the Wigner function. A Gaussian ansatz then allows for reducing an infinite amount of differential equations (one for each entry of the density matrix) to four coupled, non-linear differential equations. This procedure is outlined in App. D. Here we focus on the long-time limit, where any transient behavior due to the initial state is negligible. The probabilistic first-law violations that we find can thus not directly be linked to the energy-time uncertainty relation.

## 6.1  Distributions of heat and work

From Eq. (50), we can derive a cumulant generating function for the joint fluctuations of heat and work in the long-time limit

$$\frac{\mathcal{S}(\chi,\lambda)}{t} = \frac{\kappa_h + \kappa_c}{2} - \frac{1}{2}\sqrt{\kappa_h^2 + \kappa_c^2 - 2g_\lambda^2 + 2\sqrt{[g_\lambda^2 + \kappa_h\kappa_c]^2 - 4g_\lambda^2\kappa_h\kappa_c\Psi(\chi,\lambda)}}, \qquad (57)$$

where we abbreviated $g_\lambda^2 = g^2(4 + \lambda^2)$ and introduced the function

$$\Psi(\chi,\lambda) = n_B^h(n_B^c + 1)\left(e^{-i\chi}\frac{1 - i\lambda/2}{1 + i\lambda/2} - 1\right) + n_B^c(n_B^h + 1)\left(e^{i\chi}\frac{1 + i\lambda/2}{1 - i\lambda/2} - 1\right). \qquad (58)$$

The quasi-probability distribution for heat and work is obtained from the cumulant generating function through Eq. (52) and illustrated in Fig. 2. This distribution exhibits two striking features:

  I  The first law of thermodynamics can be violated probabilistically.

 II  The distribution takes on negative values.

The first law violations are visualized by finite values of $P(q, w)$ for $q \neq w$ and are a direct consequence of the fact that $S(\chi,\lambda) \neq S(\chi + \lambda)$. It is instructive to inspect the function $\Psi(\chi,\lambda)$. The first (second) term in Eq. (58) corresponds to photons going from hot to cold (cold to hot). The terms in the brackets show how these photons contribute to heat and work. In general, their contributions to heat and work are different. Only for small $\lambda$ do these *counting terms* reduce to $\exp[\mp i(\chi + \lambda)] - 1$. Furthermore, Eq. (57) shows an additional influence on work fluctuations arising from a rescaling of the interaction strength $g^2 \to g_\lambda^2 = g^2(4 + \lambda^2)$. We note that while $q = w$ is not enforced on the level of (quasi-)probabilities, the first law still holds on average as we find $\langle\langle q \rangle\rangle = \langle\langle w \rangle\rangle$.

The negative values of the quasi-probability distribution reflect the fact that it is impossible to measure heat and work simultaneously without affecting these quantities by measurement backaction. Such negativities have been found before in the charge transfer between superconductors [37].

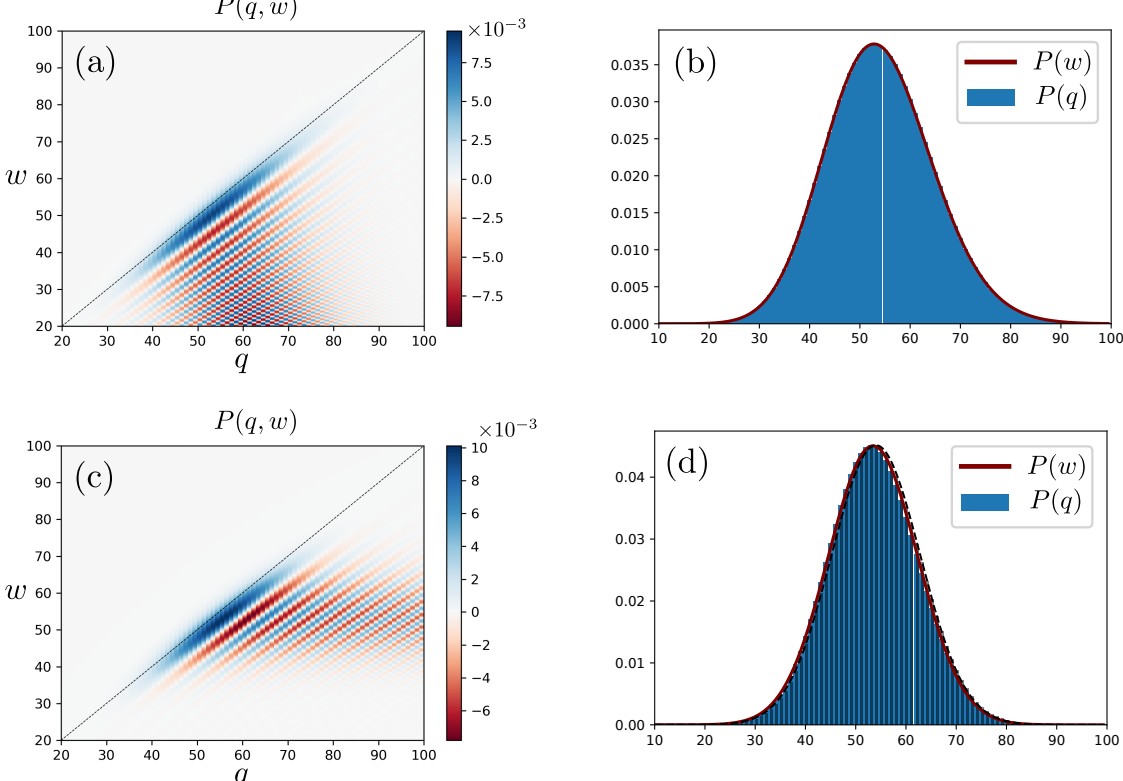

Figure 2: Joint distribution of heat and work together with its marginals for $g/\kappa = 1$ [(a) and (b)] and $g/\kappa = 0.05$ [(c) and (d)]. Here $\kappa \equiv \kappa_c = \kappa_h$. The joint distribution for heat ($q$) and work ($w$) [(a) and (c)] features negative values, as well as probabilistic first law violations (non-negative values away from the diagonal dashed line). As discussed in the main text, these features arise from quantum coherence, preventing a back-action-free and precise measurement of work. The marginal distributions of heat and work [(b) and (d)] are described by a discrete and continuous distribution respectively. This is a consequence of the fact that photons can be counted one by one while Cooper pairs cannot. The dashed line and dark blue bars in (d) show the analytic expressions obtained for $g \ll \kappa$. For large $g$ ($\sim 20\kappa$), the distributions look qualitatively the same as for $g \sim \kappa$ (not shown). Parameters: $n_B^h = 1$, $n_B^c = 0.1$, $gt = \kappa t = 150$ [(a) and (b)], $gt = 600$, $\kappa t = 12'000$ [(c) and (d)], ensuring the same mean values in panels (a) and (c).

Both features I and II become particularly apparent in the distribution describing the difference of work and heat defined as

$$P(\Delta) = \sum_{q=-\infty}^{\infty} P(q, q+\Delta) = \int_{-\infty}^{\infty} \frac{d\lambda}{2\pi} e^{\mathcal{S}(-\lambda,\lambda)+i\lambda\Delta}. \tag{59}$$

This distribution is illustrated in Fig. 3. It exhibits oscillations and negative values. The fact that it takes on non-zero values for $\Delta = w - q \neq 0$ implies the presence of probabilistic first law violations. Note that the condition for probabilistic violations of the first law can be cast into [cf. Eq. (56)]

$$\exists\; k \;\; \text{s.t.} \;\; \langle\langle \Delta^k \rangle\rangle \neq 0. \tag{60}$$

We note that the probabilistic violations of the first law occur almost exclusively for $w < q$. As shown in Sec. 6.5, this implies that measuring a work value less than the expended heat is vastly more probable than measuring work that exceeds the heat input.

## 6.2 Cumulants

For understanding both features I and II, it is instructive to consider the cumulants that follow from Eq. (57). The averages read (see also, [18,75])

$$\frac{\langle\langle q\rangle\rangle}{t} = \frac{\langle\langle w\rangle\rangle}{t} = \frac{4g^2\kappa_h\kappa_c(n_B^h - n_B^c)}{(4g^2 + \kappa_h\kappa_c)(\kappa_h + \kappa_c)}, \tag{61}$$

for the (co-)variances, we find

$$\langle\langle q^2\rangle\rangle = \langle\langle w^2\rangle\rangle = \langle\langle qw\rangle\rangle = \langle\langle q\rangle\rangle \coth\left(\frac{\beta_c\Omega_c - \beta_h\Omega_h}{2}\right) + 2\frac{\langle\langle q\rangle\rangle^2}{t}\frac{(\kappa_c + \kappa_h)^2 + (4g^2 + \kappa_c\kappa_h)}{(\kappa_c + \kappa_h)(4g^2 + \kappa_c\kappa_h)}, \tag{62}$$

and the third order cumulants read

$$\langle\langle q^3\rangle\rangle = \langle\langle q^2 w\rangle\rangle = \langle\langle q\rangle\rangle\left[1 + 6\frac{\langle\langle q^2\rangle\rangle}{t}\frac{(\kappa_c + \kappa_h)^2 + (4g^2 + \kappa_c\kappa_h)}{(\kappa_c + \kappa_h)(4g^2 + \kappa_c\kappa_h)} - \frac{12\langle\langle q\rangle\rangle^2/t^2}{4g^2 + \kappa_c\kappa_h}\right], \tag{63}$$

$$\langle\langle qw^2\rangle\rangle = \langle\langle q^3\rangle\rangle - \frac{1}{2}\langle\langle q\rangle\rangle\frac{\kappa_c\kappa_h}{4g^2 + \kappa_c\kappa_h}, \qquad \langle\langle w^3\rangle\rangle = \langle\langle q^3\rangle\rangle + \langle\langle q\rangle\rangle\frac{2g^2 - \kappa_c\kappa_h}{4g^2 + \kappa_c\kappa_h}.$$

We thus find that for the cumulants up to second order, the distribution behaves as if $q = w$ (in agreement with Ref. [21]). However, starting from the third order cumulants, this is no longer the case. In particular, from Eq. (59), we find that the first non-vanishing cumulants for $\Delta = w - q$ are

$$\langle\langle\Delta^3\rangle\rangle = \frac{1}{2}\langle\langle q\rangle\rangle, \qquad \langle\langle\Delta^5\rangle\rangle = \langle\langle q\rangle\rangle\frac{6g^2 - \kappa_c\kappa_h}{4g^2\kappa_c\kappa_h}. \tag{64}$$

We note that for higher cumulants, also even cumulants are non-vanishing. The distribution $P(\Delta)$ thus has both a vanishing mean and a vanishing variance. For any non-negative distribution, vanishing mean and variance imply that all cumulants vanish (i.e., the distribution is the Dirac delta distribution). The finite higher cumulants in Eq. (64) thus imply that $P(\Delta)$ takes on negative values *and* exhibits probabilistic first law violations [cf. Eq. (60)]. When the first law holds up to the second order cumulants, probabilistic first law violations are thus intimately linked to negative quasi-probabilities.

Since higher cumulants capture the behavior of the tails of a distribution, one can usually reproduce the characteristic features by only keeping the lowest cumulants in the generating function. We thus approximate

$$P(\Delta) \approx \int_{-\infty}^{\infty} \frac{d\lambda}{2\pi} e^{i\left(\frac{\langle\langle q\rangle\rangle}{12}\lambda^3 + \lambda\Delta\right)} = \left(\frac{4}{|\langle\langle q\rangle\rangle|}\right)^{\frac{1}{3}} \text{Ai}\left[\frac{\Delta}{(\langle\langle q\rangle\rangle/4)^{\frac{1}{3}}}\right], \tag{65}$$

where the Airy function of the first kind is defined as

$$\text{Ai}(x) = \int_{-\infty}^{\infty} dt\, e^{i\left(\frac{t^3}{3} + xt\right)}. \tag{66}$$

Equation (65) captures the behavior of $P(\Delta)$ well for small values of $\Delta$, cf. Fig. 3.

## 6.3 Limiting cases

To get a better analytical understanding of the heat-to-work conversion process, it is instructive to consider the two limiting cases $\kappa_c, \kappa_h \ll g$ and $g \ll \kappa_c, \kappa_h$. For $\kappa_c, \kappa_h \ll g$, we find

$$\frac{\mathcal{S}(\chi,\lambda)}{t} = \frac{\kappa_h + \kappa_c}{2}\left[1 - \sqrt{1 - \frac{4\kappa_h\kappa_c}{(\kappa_h + \kappa_c)^2}\Psi(\chi,\lambda)}\right]. \tag{67}$$

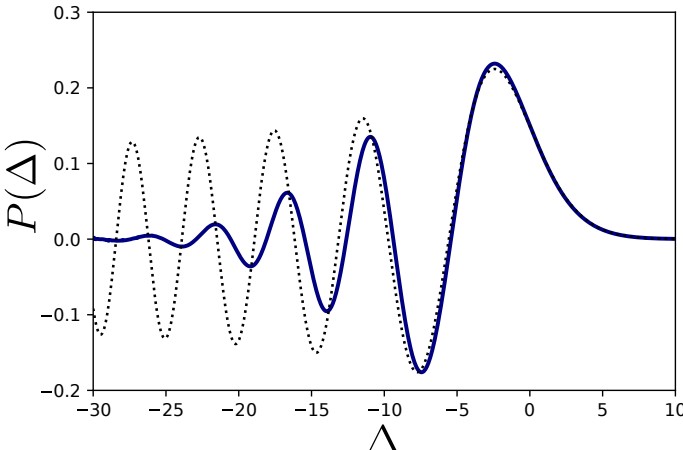

Figure 3: Distribution for the difference of work and heat $\Delta = w - q$. Having a vanishing mean and variance, this distribution is bound to have negative values if probabilistic violations of the first law are present (i.e., if it is not equal to a Dirac delta). The dotted line shows the Airy function in Eq. (66), exhibiting good agreement with $P(\Delta)$ for small values of $\Delta$. Parameters: $n_B^h = 1$, $n_B^c = 0.1$, $gt = \kappa t = 150$.

For $\lambda = 0$, this equation reduces to the cumulant generating function for a single resonator coupled to two baths, cf. Eq. (F14) in Ref. [76]. Due to the strong coupling between the resonators, they act similarly to one single resonator. Note however that the frequencies associated to the two reservoirs are different ($\Omega_c \neq \Omega_h$), reflecting the fact that photons change their energy when going from one resonator to the other. The analogy with a single resonator should thus be treated with care.

In the opposite limit, $g \ll \kappa_c, \kappa_h$, we find

$$\frac{\mathcal{S}(\chi, \lambda)}{t} = \frac{g^2(4 + \lambda^2)}{\kappa_h + \kappa_c} \Psi(\chi, \lambda). \tag{68}$$

In this limit, we obtain the marginals analytically. For the heat distribution, we find a bidirectional Poissonian with the distribution

$$P(q) = e^{-(\Gamma_{ch} + \Gamma_{hc})t} \left( \frac{\Gamma_{ch}}{\Gamma_{hc}} \right)^{\frac{q}{2}} I_q \left[ 2t \sqrt{\Gamma_{ch}\Gamma_{hc}} \right], \tag{69}$$

where $I_q$ denotes the modified Bessel function of the first kind. The rate for photons to go from reservoir $\alpha$ to reservoir $\beta$ is given by

$$\Gamma_{\beta\alpha} = \frac{4g^2}{\kappa_c + \kappa_h} n_B^\alpha (n_B^\beta + 1). \tag{70}$$

The moments of this distribution are particularly simple and read

$$\langle\langle q^k \rangle\rangle / t = \begin{cases} \Gamma_{ch} - \Gamma_{hc} & \text{for } k \text{ odd}, \\ \Gamma_{ch} + \Gamma_{hc} & \text{for } k \text{ even}. \end{cases} \tag{71}$$

Such a bi-directional Poissonian describes particles being partitioned at a single junction. In the present case, the Josephson junction provides a bottleneck for the photons, making it the only junction that is relevant for transport statistics.

For the work distribution, we find a Gaussian distribution with the same mean and average as for the heat

$$P(w) = \frac{1}{\sqrt{2\pi\langle\langle q^2 \rangle\rangle}} e^{-\frac{(w - \langle\langle q \rangle\rangle)^2}{2\langle\langle q^2 \rangle\rangle}}. \tag{72}$$

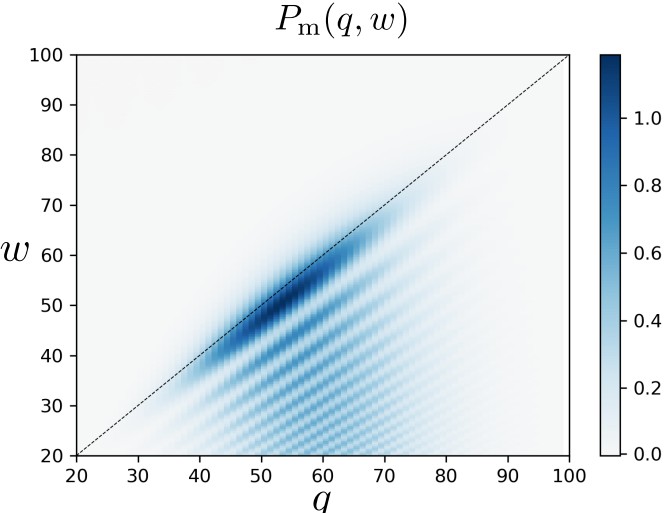

Figure 4: Distribution describing a measurement of heat and work. While heat is assumed to be measured perfectly (e.g., by monitoring the photons that enter and leave the system), the measurement of work is described by Eq. (8). The measurement strength is quantified by $\sigma = 0.35$. The rest of the parameters are the same as in Fig. 2 (a).

## 6.4 Out-of-equilibrium relations

Having access to the analytic expression of the cumulant generating function [cf. Eq. (57)], we may verify out-of-equilibrium relations that are expected to hold for the present scenario. In particular, we consider the fluctuation theorem [3] as well as the thermodynamic uncertainty relation (TUR) [77].

From the symmetry

$$\mathcal{S}(\chi, \lambda) = \mathcal{S}(\chi + i\beta_h \Omega_h - i\beta_c \Omega_c, -\lambda), \tag{73}$$

a fluctuation theorem follows

$$\frac{P(-q, -w)}{P(q, w)} = e^{-q(\beta_c \Omega_c - \beta_h \Omega_h)}. \tag{74}$$

While this implies a fluctuation theorem for heat, see Eq. (43), no simple fluctuation theorem for work can be derived from Eq. (74). Indeed, derivations for work fluctuation theorems usually rely on the first law in order to relate work to entropy. Not surprisingly, this approach breaks down in the presence of probabilistic first law violations.

From the second cumulant given in Eq. (62), we find that the TUR is obeyed in our system

$$\frac{\langle\langle q^2 \rangle\rangle}{\langle\langle q \rangle\rangle^2} \geq \frac{2k_B}{\langle\langle \Sigma \rangle\rangle}, \tag{75}$$

where the average entropy production reads $\langle\langle \Sigma \rangle\rangle = \langle\langle q \rangle\rangle(\beta_c \Omega_c - \beta_h \Omega_h)$. Equation (75) can be proven by noting that the second term in the variance [cf. Eq. (62)] is strictly positive, and by using the inequality $x \coth(x) \geq 1$. Since heat fluctuations are equal to work fluctuations in our system, the TUR implies a direct trade-off between power, efficiency, and power fluctuations [78] for the present quantum heat engine. The validity of the TUR is in agreement with Ref. [79], where it was shown that the TUR is valid for harmonic oscillator junctions.

## 6.5 The joint probability distribution for measuring heat and work

For a joint measurement of heat and work, we consider a detector with a Gaussian Wigner function [cf. Eq. (84)]

$$\mathcal{W}(x,p) = \frac{1}{2\pi\sigma_x\sigma_p} e^{-\frac{x^2}{2\sigma_x^2}} e^{-\frac{p^2}{2\sigma_p^2}} . \tag{76}$$

To ensure an optimal trade-off between measurement imprecision and back-action, we choose $\sigma_x\sigma_p = 1/2$, saturating the uncertainty principle, and define $\sigma \equiv \sigma_x/s$ which is the only remaining free parameter describing the influence of the detector on the joint distribution. We note that for this optimal trade-off between measurement imprecision and back-action, the detector generally has to be in a pure state. The energetic cost of preparing such a state (which diverges due to the third law of thermodynamics [80, 81]) is not taken into account here. We stress however that this does not hamper our conclusions for weak measurements, which only rely on a Gaussian distribution for the detector variable $\hat{r}$.

Coupling a detector to the system in general modifies the energy balance even on average, see App. B. This may act as a resource, fueling engines and refrigerators [49, 82–84]. We note however that the energy exchanged with the detector does not restore the first law for the measured distribution, since measuring this energy would require an additional detector which would face the exact same problems.

The probability distribution for the outcomes of a joint measurement of work and heat is shown in Fig. 4. While measurement imprecision and back-action alter the quasi-probability distribution rendering it non-negative, similarities are clearly visible [cf. Fig. 2]. In particular, the oscillatory behavior which results from the coherent nature of the heat-to-work conversion process remains visible. Furthermore, the measured distribution exhibits probabilistic violations of the first law, just as the underlying quasi-probability distribution. As a consequence, a work value that is considerably smaller than the heat input may be measured.

First law violations in the measurement are to be expected because neither back-action nor imprecision are generally able to restore the first law on the level of probabilities. While there are some instances where back-action may ensure the validity of the first law, this is not generally the case and it may even alter the energy balance on average. In addition, measurement imprecision introduces randomness to the measurement outcomes in a way that is unrelated to energy conservation.

## 7 Conclusions & Outlook

Fluctuations challenge the laws of thermodynamics. While they hold on average, it is well established that the second law of thermodynamics can be probabilistically violated. Here we showed that this holds true for the first law as well, in the presence of quantum fluctuations. An imprecision vs backaction tradeoff will in general prevent the first law of thermodynamics to be applicable for single experimental runs, when heat, work, and internal energy are determined by separate measurements. In order to determine the statistics of these quantities in the absence of any measurement, a weak measurement may be employed and sub-sequentially corrected for measurement imprecision. This procedure results in a quasi-probability distribution that becomes non-negative and respects the first law of thermodynamics in a classical regime. For quantum systems, this QPD does in general not respect the first law.

To illustrate these probabilistic violations of the first law, we provided a detailed investigation into the joint fluctuations of heat and work in a quantum heat engine. In this device, heat fluctuations behave classically due to the weak coupling between system and bath. At strong coupling, a thermodynamically consistent definition of internal energy and heat is a

subject of debate [40–42], further challenging the validity of the first law on the level of probabilities. Additionally, on time-scales shorter than the bath correlation time, the time-energy uncertainty relation may render energy conversion inapplicable [85, 86].

In agreement with previous works [87, 88], our results imply that conservation laws, such as energy conservation, do not enjoy the same impact in quantum systems as they do in classical systems. The reason for this is that when dividing a conserved quantity into parts, a Heisenberg uncertainty relation may restrict our simultaneous knowledge of these parts. The quasi-probability distribution used to describe work fluctuations here is a member of a larger family, which have been called Keldysh quasi-probability distributions [32]. Another prominent member is the Wigner function and generalizations thereof [89]. Different Keldysh quasi-probability distributions may shed light onto the impact of different conservation laws on measurable but non-commuting quantities.

Our results open up several interesting avenues: Equation (74) implies that a usual Crooks fluctuation theorem for work cannot be derived when the first law of thermodynamics does not hold on the level of the distribution. Nevertheless, fluctuation theorems including quantum corrections have been derived [90, 91]. These quantum corrections may have a connection to the probabilistic violations of the first law investigated here.

A further interesting avenue to pursue is provided by potential links between probabilistic violations of the first law and the energy-time uncertainty relation. Such links are expected to become particularly important in the short-time regime not considered in the present manuscript.

Another open question concerns the relation between first law violations and negative values in the Keldysh quasi-probability distribution. Such negative values have recently been shown to be a necessary and sufficient condition for ruling out a classical description of the measurement outcomes [92]. Here we showed that if the first law holds not only for the first but also for the second cumulants, then first law violations are accompanied by negative quasi-probabilities. While the first law is only a statement about average values, we may speculate that it also generally holds for the second cumulants. If this were true, it would strengthen the link between probabilistic violations of the first law and negative quasi-probabilities.

Probabilistic violations of the second law are by now well understood, in particular because of fluctuation theorems which are recognized as the generalization of the second law to fluctuating systems. By demonstrating probabilistic first law violations, our results may open a quest for finding a generalization of the first law that includes quantum fluctuations. At this point, we may only speculate how such a generalization may look like and what its potential impact may be.

## Acknowledgments

We acknowledge fruitful discussions with the audience from the Quarantine Thermo seminar series as well as the QTD2020 conference, in particular with P. Strasberg and C. Elouard. We further thank P. Hänggi, G. Haack, M. Huber, A. Jordan, G. Manzano, M. Perarnau-Llobet, and R. Sánchez for valuable feedback on the manuscript. This work was supported by the Swedish Research Council. T.K. acknowledges the Knut and Alice Walenberg Foundation (KAW) (project 2016.0089). P.P.P. acknowledges funding from the European Union's Horizon 2020 research and innovation programme under the Marie Skłodowska-Curie Grant Agreement No. 796700, from the Swedish Research Council (Starting Grant 2020-03362), as well as from the Swiss National Science Foundation (Eccellenza Professorial Fellowship PCEFP2_194268).

# A  The general scenario

In this Appendix, we provide details for the general scenario discussed in Sec. 2. There we introduced measurements for internal work, heat, and internal energy changes. A joint measurement of these quantities provides outcomes $W$, $Q$, and $\Delta U$ with probability

$$P_{\mathrm{m}}(Q,W,\Delta U) = s\sum_{E_{\mathrm{B}}^{\tau},E_{\mathrm{B}}^{0}}\sum_{E^{\tau},E^{0}}\langle E^{\tau},E_{\mathrm{B}}^{\tau},sW|\hat{U}_{\mathrm{m}}(\tau)|E^{0},E_{\mathrm{B}}^{0}\rangle\langle E^{0},E_{\mathrm{B}}^{0}|\otimes\hat{\rho}_{\mathrm{d}}\hat{U}_{\mathrm{m}}^{\dagger}(\tau)|E^{\tau},E_{\mathrm{B}}^{\tau},sW\rangle$$
$$\times\langle E^{0},E_{\mathrm{B}}^{0}|\hat{\rho}_{\mathrm{tot}}(0)|E^{0},E_{\mathrm{B}}^{0}\rangle\delta\left(Q-E_{\mathrm{B}}^{0}+E_{\mathrm{B}}^{\tau}\right)\delta\left(\Delta U-E^{\tau}+E^{0}\right), \tag{77}$$

where the time-evolution includes the detector

$$\hat{U}_{\mathrm{m}}(\tau) = \mathcal{T}e^{-i\int_{0}^{\tau}dt[\hat{H}(t)+\hat{H}_{\mathrm{B}}+\hat{V}+s\hat{P}(t)\hat{\pi}]}, \tag{78}$$

with the time-ordering operator $\mathcal{T}$. Furthermore, $|E_{\mathrm{B}}^{t},E^{t},r\rangle$ is a joint eigenstate of $\hat{H}_{\mathrm{B}}$, $\hat{H}(t)$, and the position operator of the detector $\hat{r}$. The initial density matrix of the detector is denoted by $\hat{\rho}_{\mathrm{d}}$. Fourier transforming Eq. (77), we obtain the moment generating function

$$\Lambda_{\mathrm{m}}(\chi,\lambda,\xi) = \int_{-\infty}^{\infty}dQ\,dW\,d\Delta U\,e^{-i\lambda W-i\chi Q-i\xi\Delta U}P_{\mathrm{m}}(Q,W,\Delta U)$$
$$= \int_{-\infty}^{\infty}dr\,e^{-i\lambda r/s}\mathrm{Tr}\left\{|r\rangle\langle r|e^{i\chi\hat{H}_{\mathrm{B}}-i\xi\hat{H}(\tau)}\hat{U}_{\mathrm{m}}(\tau)e^{-i\chi\hat{H}_{\mathrm{B}}+i\xi\hat{H}(0)}\hat{\rho}_{\mathrm{tot}}(0)\otimes\hat{\rho}_{\mathrm{d}}\hat{U}_{\mathrm{m}}^{\dagger}(\tau)\right\}$$
$$= \int d\gamma\frac{1}{s}\langle(\gamma+\lambda/2)/s|\hat{\rho}_{\mathrm{d}}|(\gamma-\lambda/2)/s\rangle\Lambda_{\gamma}(\chi,\lambda,\xi), \tag{79}$$

where the states that sandwich $\hat{\rho}_{\mathrm{d}}$ in the last row are eigenstates of the momentum operator $\hat{\pi}$. Furthermore, we assumed $\hat{\rho}_{\mathrm{tot}}(0)$ to commute both with $\hat{H}(0)$ as well as with $\hat{H}_{\mathrm{B}}$ and we introduced the moment generating function of the QPD

$$\Lambda_{\gamma}(\chi,\lambda,\xi) = \mathrm{Tr}\left\{e^{i\chi\hat{H}_{\mathrm{B}}-i\xi\hat{H}(\tau)}\hat{U}(\tau;\gamma+\lambda/2)e^{-i\chi\hat{H}_{\mathrm{B}}+i\xi\hat{H}(0)}\hat{\rho}_{\mathrm{tot}}(0)\hat{U}^{\dagger}(\tau;\gamma-\lambda/2)\right\}, \tag{80}$$

with the modified time-evolution operator

$$\hat{U}(\tau;z) = \mathcal{T}e^{-i\int_{0}^{\tau}dt[\hat{H}(t)+\hat{H}_{\mathrm{B}}+\hat{V}+z\hat{P}(t)]}, \tag{81}$$

which only acts on the Hilbert space of system and bath (not including the detector).

From Eq. (79), using the identities

$$P_{\gamma}(Q,W,\Delta U) = \int_{-\infty}^{\infty}\frac{d\lambda}{2\pi}\frac{d\chi}{2\pi}\frac{d\xi}{2\pi}e^{i\lambda W+i\chi Q+i\xi\Delta U}\Lambda_{\gamma}(\chi,\lambda,\xi), \tag{82}$$

and

$$\frac{1}{s}\langle(\gamma+\lambda/2)/s|\hat{\rho}_{\mathrm{d}}|(\gamma-\lambda/2)/s\rangle = \int_{-\infty}^{\infty}dW\,e^{-i\lambda W}\mathcal{W}(Ws,\gamma/s), \tag{83}$$

together with the convolution theorem, we find

$$P_{\mathrm{m}}(Q,W,\Delta U) = \int_{-\infty}^{\infty}d\gamma\int_{-\infty}^{\infty}dW'\mathcal{W}([W-W']s,\gamma/s)P_{\gamma}(Q,W',\Delta U). \tag{84}$$

As discussed in the main text, the probabilities describing the measurement can be written as a convolution of the QPD and the Wigner function of the detector used for measuring work.

## A.1 The classical regime

In the classical regime, we make the assumptions $[\hat{H}(t), \hat{H}(t')] = 0$ as well as $[\hat{V}, \hat{H}(t) + \hat{H}_{\mathrm{B}}] = 0$. From these assumptions, it follows that $[P(t), \hat{H}_{\mathrm{tot}}(t')] = 0$ ($[P(t), \hat{V}] = 0$ follows from the fact that $[H(t), \hat{V}] = -[H_{\mathrm{B}}, \hat{V}]$ is time-independent). We can then write

$$\hat{U}(\tau; z) = \hat{U}(\tau) e^{-iz \int_0^\tau dt \hat{P}(t)} = e^{iz[\hat{H}(\tau) + \hat{H}_{\mathrm{B}}]} \hat{U}(\tau) e^{-iz[\hat{H}(0) + \hat{H}_{\mathrm{B}}]}. \tag{85}$$

In this regime, Eq. (80) reduces to

$$\Lambda_\gamma(\chi, \lambda, \xi) = \Lambda_0(\chi + \lambda, 0, \xi - \lambda), \tag{86}$$

which implies $P_\gamma(Q, W, \Delta U) = P(Q, \Delta U)\delta(W - Q + \Delta U)$, where $P(Q, \Delta U)$ describes a joint measurement of heat and internal energy change, in the absence of a detector for measuring work. In the classical regime, backaction thus becomes irrelevant and the QPD respects the first law of thermodynamics on the level of probabilities.

## A.2 Weak measurements

For a weak measurement, when $s \to 0$, a finite momentum support of the detector Wigner function implies that we can make the replacement $\mathcal{W}([W - W']s, \gamma/s) \to p_{\mathrm{d}}(W - W')\delta(\gamma)$, where

$$p_{\mathrm{d}}(W - W') = s \int dp \, \mathcal{W}([W - W']s, p). \tag{87}$$

The integral over $\gamma$ can then trivially be executed in Eq. (84) and from the convolution theorem, we find a simple relation between the measured cumulants and the cumulants of the QPD

$$\langle\langle Q^j W^k \Delta U^l \rangle\rangle_{\mathrm{m}} = c_k + \langle\langle Q^j W^k \Delta U^l \rangle\rangle, \tag{88}$$

where $c_k$ denotes the $k$-th cumulant of $p_{\mathrm{d}}(W)$. For a detector with a Gaussian initial position distribution, we recover Eq. (16).

# B The first law in the presence of the detector

The coupling to the detector represents a time-dependent contribution to the Hamiltonian, cf. Eq. (7). This term modifies the energy balance resulting in a modified first law

$$\langle \dot{U} \rangle = -\langle \dot{W} \rangle - \langle \dot{W}_{\mathrm{d}} \rangle + \langle \dot{Q} \rangle, \tag{89}$$

where we used the subscript d to label the power provided by the detector. The contribution from the detector to the first law reads

$$\langle \dot{W}_{\mathrm{d}} \rangle = -\mathrm{Tr}\{\hat{\rho} \, \partial_t \hat{H}_{\mathrm{m}}\}. \tag{90}$$

Since the detector couples to the power operator in the *absence* of the detector itself, it only measures $W$, which is no longer the total work. This is a general issue: whenever a time-dependent power operator is being measured, the measurement Hamiltonian will necessarily also be time-dependent, providing an additional contribution to the total power. With a linear detector, it is impossible to include this contribution in the measurement. If we however consider weak measurements, then $W_{\mathrm{d}}$ can safely be neglected. This can be seen by noting that the Hamiltonian describing the coupling to the detector is proportional to $s\hat{\pi}$. As the Hamiltonian commutes with $\hat{\pi}$, the momentum values of the detector much larger than $\sigma_p$

are negligible, cf. Eq. (76), and the power provided by the detector will be proportional to a factor smaller than $s\sigma_p = 1/(2\sigma)$, which vanishes for weak measurements.

Just as in the absence of the detector, Eq. (89) holds for average values and does not take over to fluctuations. Indeed, when adding a second detector to measure $W_\mathrm{d}$, the energy balance gets modified again. Including the energetics of the detector does therefore not salvage the first law of thermodynamics beyond average values.

## C  The long-time limit

In the main text, we argued that the internal energy of the system may be neglected in the long time limit. We further stated that the fluctuations of heat are completely determined by a single variable $q$, describing the net number of photons that went from the hot to the cold reservoir. Here we prove these two statements. We start with the moment generating function for heat, work, and internal energy changes

$$e^{\mathcal{S}(\boldsymbol{\chi},\lambda,\xi)} = \mathrm{Tr}\left\{e^{-i\xi\hat{H}(\tau)}e^{\mathcal{L}(\boldsymbol{\chi},\lambda)t}e^{i\xi\hat{H}(0)}\hat{\rho}\right\}, \tag{91}$$

where the Liouvillian $\mathcal{L}(\boldsymbol{\chi},\lambda)$ is determined by the right-hand side of Eq. (50). The counting field $\xi$ corresponds to the internal energy fluctuations. We may re-write Eq. (91) using the spectral decomposition of the Liouvillian

$$e^{\mathcal{S}} = \sum_i e^{\nu_i t}\mathrm{Tr}\left\{e^{-i\xi\hat{H}(\tau)}\mathcal{P}_i e^{i\xi\hat{H}(0)}\hat{\rho}\right\}. \tag{92}$$

The eigenvalues of the Liouvillian are denoted by $\nu_i$ and the projectors $\mathcal{P}_i$ depend on the eigenstates of the Liouvillian. In the long-time limit, the sum is dominated by the eigenvalue with the largest real part, denoted $\nu_\mathrm{max}$, and we find

$$\mathcal{S}(\boldsymbol{\chi},\lambda,\xi) = \nu_\mathrm{max}(\boldsymbol{\chi},\lambda)t + \mathcal{S}_0(\boldsymbol{\chi},\lambda,\xi), \tag{93}$$

where the time-independent term is given by

$$\mathcal{S}_0 = \ln\mathrm{Tr}\left\{e^{-i\xi\hat{H}(\tau)}\mathcal{P}_\mathrm{max}(\boldsymbol{\chi},\lambda)e^{i\xi\hat{H}(0)}\hat{\rho}\right\}. \tag{94}$$

In the long-time limit, we may safely drop the term $\mathcal{S}_0$ in the cumulant generating function. In particular, for the cumulants we find

$$\langle\langle(Q-W-\Delta U)^k\rangle\rangle/t = \langle\langle(Q-W)^k\rangle\rangle/t + \mathcal{O}(1/t). \tag{95}$$

In particular, this implies that the first law may not be recovered by taking into account the internal energy fluctuations, as these scale differently with time.

We now show that the long-time limit, the statistics of heat is fully determined by a single counting field, i.e., $P(\boldsymbol{q}) \propto \delta_{q_h,-q_c}$. To this end, we introduce the unitary superoperator

$$\mathcal{U}\hat{A} = e^{-i\frac{\chi_c}{2}(\hat{a}_c^\dagger\hat{a}_c + \hat{a}_h^\dagger\hat{a}_h)}\hat{A}e^{-i\frac{\chi_c}{2}(\hat{a}_c^\dagger\hat{a}_c + \hat{a}_h^\dagger\hat{a}_h)}. \tag{96}$$

To show that this superoperator is unitary, we note that its adjoint, $\mathcal{U}^\dagger$, is defined by

$$\langle\mathcal{U}^\dagger\hat{B},\hat{A}\rangle = \langle\hat{B},\mathcal{U}\hat{A}\rangle, \tag{97}$$

where we introduced the inner product $\langle\hat{B},\hat{A}\rangle = \mathrm{Tr}\{\hat{B}^\dagger\hat{A}\}$. We find

$$\mathcal{U}^\dagger\hat{A} = e^{i\frac{\chi_c}{2}(\hat{a}_c^\dagger\hat{a}_c + \hat{a}_h^\dagger\hat{a}_h)}\hat{A}e^{i\frac{\chi_c}{2}(\hat{a}_c^\dagger\hat{a}_c + \hat{a}_h^\dagger\hat{a}_h)}, \tag{98}$$

and it follows that $\mathcal{U}^\dagger \mathcal{U} = 1$.

We now consider the Liouvillian, setting $\lambda = 0$ for simplicity (the proof also holds for $\lambda \neq 0$)

$$\tilde{\mathcal{L}}(\chi_h - \chi_c)\hat{\rho} = \mathcal{U}^\dagger \mathcal{L}(\boldsymbol{\chi})\mathcal{U}\hat{\rho} = -i[\hat{H}, \hat{\rho}] + \mathcal{L}_h^{\chi_h - \chi_c}\hat{\rho} + \mathcal{L}_c\hat{\rho}. \tag{99}$$

As $\tilde{\mathcal{L}}$ only depends on the difference of the counting fields, so do its eigenvalues. Since a unitary transformation leaves the eigenvalues invariant, the same holds for the eigenvalues of the original Liouvillian $\mathcal{L}(\boldsymbol{\chi})$. In the long-time limit, the cumulant generating function thus only depends on the difference in counting fields, cf. Eq. (93). The relation $P(\boldsymbol{q}) \propto \delta_{q_h, -q_c}$ then follows directly from Eq. (39). We note that for short times, the cumulant generating function also depends on the eigenvectors of the Liouvillian and therefore explicitly depends on both $\chi_h$ and $\chi_c$.

## D  The Wigner function approach

In this appendix, we show how the distribution $P(\boldsymbol{q}, w; \gamma)$ can be calculated. All results shown in the main text follow from this distribution. As shown in Sec. 5, the cumulant generating function of this distribution can be written as $\mathcal{S} = \ln[\text{Tr}\{\hat{\rho}\}]$, where $\hat{\rho}$ is governed by the master equation

$$\partial_t \hat{\rho} = -i[\hat{H} + \gamma\hat{I}, \hat{\rho}] - i\frac{\lambda}{2}\{\hat{I}, \hat{\rho}\} + \mathcal{L}_h^{\chi_h}\hat{\rho} + \mathcal{L}_c^{\chi_c}\hat{\rho}. \tag{100}$$

Here the Hamiltonian is given in Eq. (20), the current operator in Eq. (30), and the superoperators responsible for dissipation in Eq. (37). To get rid of the time-dependence, we perform a unitary transformation with

$$\hat{U} = e^{i\Omega_h t \hat{a}_h^\dagger \hat{a}_h + i\Omega_c t \hat{a}_c^\dagger \hat{a}_c}. \tag{101}$$

We then cast Eq. (100) into an equation of motion for the Wigner function

$$W(\boldsymbol{r}_c, \boldsymbol{r}_h) = \int \frac{dy_c}{2\pi}\frac{dy_h}{2\pi} e^{i(p_c y_c + p_h y_h)} \left\langle x_h + \frac{y_h}{2}, x_c + \frac{y_c}{2} \middle| \hat{\rho} \middle| x_c + \frac{y_c}{2}, x_h + \frac{y_h}{2} \right\rangle, \tag{102}$$

where $r_\alpha = (x_\alpha, p_\alpha)$ and $\hat{x}_\alpha |x_h, x_c\rangle = x_\alpha |x_h, x_c\rangle$, with the quadrature operators defined by $\hat{a}_\alpha = (\hat{x}_\alpha + i\hat{p}_\alpha)/\sqrt{2}$. Introducing the nabla operators $\boldsymbol{\nabla}_\alpha = (\partial_{x_\alpha}, \partial_{p_\alpha})$, we may write the equation of motion for the Wigner function as

$$\begin{aligned}\partial_t W(\boldsymbol{r}_c, \boldsymbol{r}_h) = &\bigg\{ ig\left[\boldsymbol{r}_h(\sigma_y - i\gamma)\boldsymbol{\nabla}_c + \boldsymbol{r}_c(\sigma_y + i\gamma)\boldsymbol{\nabla}_h\right] - \lambda g\left(\boldsymbol{r}_h\sigma_y\boldsymbol{r}_c - \boldsymbol{\nabla}_h\sigma_y\boldsymbol{\nabla}_c\right) \\ &+ \sum_{\alpha=c,h}\frac{\kappa_\alpha}{2}\left[\boldsymbol{\nabla}_\alpha\cdot\boldsymbol{r}_\alpha + (n_B^\alpha + 1/2)\nabla_\alpha^2 + \xi_+^\alpha(r_\alpha^2 + \nabla_\alpha^2/4) + \xi_-^\alpha(\boldsymbol{\nabla}_\alpha\cdot\boldsymbol{r}_\alpha - 1)\right]\bigg\}W(\boldsymbol{r}_c, \boldsymbol{r}_h),\end{aligned} \tag{103}$$

where $\sigma_y$ denotes the Pauli $y$-matrix and we abbreviated

$$\xi_\pm^\alpha = (n_B^\alpha + 1)\left(e^{i\chi_\alpha} - 1\right) \pm n_B^\alpha\left(e^{-i\chi_\alpha} - 1\right). \tag{104}$$

Equation (103) can be solved by the Gaussian ansatz

$$W(\boldsymbol{r}_c, \boldsymbol{r}_h) = \frac{e^{\mathcal{S}}}{(2\pi)^2\sqrt{\det\Sigma}}e^{-\frac{1}{2}(\boldsymbol{r}_c, \boldsymbol{r}_h)^T\Sigma^{-1}(\boldsymbol{r}_c, \boldsymbol{r}_h)}, \qquad \Sigma = \begin{pmatrix} \sigma_c & 0 & \sigma_{ch} & s_{ch} \\ 0 & \sigma_c & -s_{ch} & \sigma_{ch} \\ \sigma_{ch} & -s_{ch} & \sigma_h & 0 \\ s_{ch} & \sigma_{ch} & 0 & \sigma_h \end{pmatrix}. \tag{105}$$

The cumulant generating function is then determined by the coupled, non-linear differential equations

$$
\begin{aligned}
\partial_t \mathcal{S} =& \kappa_c \xi_+^c \sigma_c - \frac{\kappa_c}{2}\xi_-^c + \kappa_h \xi_+^h \sigma_h - \frac{\kappa_h}{2}\xi_-^h - 2i\lambda g s_{ch}, \\
\partial_t \sigma_c =& 2g(s_{ch} - \gamma\sigma_{ch}) + \kappa_c(n_B^c + 1/2 - \sigma_c) - 2i\lambda g\sigma_c s_{ch} \\
& + \kappa_c \xi_+^c(\sigma_c^2 + 1/4) - \kappa_c \xi_-^c \sigma_c + \kappa_h \xi_+^h(\sigma_{ch}^2 + s_{ch}^2), \\
\partial_t \sigma_h =& -2g(s_{ch} - \gamma\sigma_{ch}) + \kappa_h(n_B^h + 1/2 - \sigma_h) - 2i\lambda g\sigma_h s_{ch} \\
& + \kappa_h \xi_+^h(\sigma_h^2 + 1/4) - \kappa_h \xi_-^h \sigma_h + \kappa_c \xi_+^c(\sigma_{ch}^2 + s_{ch}^2), \\
\partial_t s_{ch} =& g(\sigma_h - \sigma_c) - \frac{\kappa_c + \kappa_h}{2}s_{ch} - i\lambda g(\sigma_c \sigma_h + s_{ch}^2 - \sigma_{ch}^2 - 1/4) \\
& + s_{ch}\left(\kappa_c \xi_+^c \sigma_c - \frac{\kappa_c}{2}\xi_-^c + \kappa_h \xi_+^h \sigma_h - \frac{\kappa_h}{2}\xi_-^h\right), \\
\partial_t \sigma_{ch} =& -\gamma g(\sigma_h - \sigma_c) - \frac{\kappa_c + \kappa_h}{2}\sigma_{ch} - 2i\lambda g\sigma_{ch}s_{ch} \\
& + \sigma_{ch}\left(\kappa_c \xi_+^c \sigma_c - \frac{\kappa_c}{2}\xi_-^c + \kappa_h \xi_+^h \sigma_h - \frac{\kappa_h}{2}\xi_-^h\right).
\end{aligned}
\tag{106}
$$

We note that for $\gamma = 0$ (i.e., in the absence of a detector), we may set $\sigma_{ch} = 0$.

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
