# Peer review of "Probabilistically Violating the First Law of Thermodynamics in a Quantum Heat Engine"

_SciPost Physics, doi:SciPost Phys. 12, 168 (2022)_

## Round 3 · Referee Report · Philipp Strasberg (Referee 2) · 2022-3-25

Report

Accessing dynamical fluctuations in quantum systems is a challenging task because, owing to measurement backaction and non-commutativity, there is no unique measurement strategy to reveal them. Instead, different approaches can give different results, and this turned out to be a particularly pressing problem in quantum stochastic thermodynamics, where one aims to define internal energy, heat, work and entropy along single "stochastic trajectories".

The manuscript elucidates this problem for a voltage biased superconducting circuit. Here, the measurement of work is realized by counting tunneling electrons using a "power operator", whereas the heat is inferred from counting the emitted and absorbed phonons. The authors find that there are violations of the stochastic first law even at steady state, i.e., the heat flow balances the work only on average, but not along single trajectories. This quantum feature, which is absent in classical stochastic thermodynamics, arises due to the fact that the two different measurement strategies are incompatible as a result of non-commutativity.

The manuscript is well structured, well written and technically sound. While it is known that measuring thermodynamic fluctuations in quantum systems is problematic, the present results are particularly intriguing because they are derived for an experimentally relevant setup with a clear physical picture behind it. I believe the manuscript will stimulate an important discussion and I hope that the here presented theory can be investigated experimentally in the future.

---

## Round 3 · Referee Report · Anonymous (Referee 3) · 2022-4-12

Report

The paper is devoted to a stochastic view of the first law of thermodynamics.
In quantum mechanics partitioning the energy to heat and work is not obvious. One can ask is there is an observable associated to these quantities?
Manny attempts have been suggested for example the two point measurement for work. This forces the system to loos coherence intefering with the dynamics. The first step therefore is to define an operational
measure for work and heat. Are these measurements compatible?
The authors suggest to measure heat by a two point measurement on the environment. For work the power is integrated on a work reservoir.
The authors study a specific heat engine composed biased superconducting circuit. As expected on average the first law of thermodynamics is obeyed.
The setup allows to account for fluctuations of the observables.
Individual realisations have a probability of violating the first law.
From a quantum viewpoint the source of the violation is that the power
"operator" employed does not commute with the energy. The authors use the tools of weak measurement and Wigner distribution function.

The paper deals with an important issue in quantum thermodynamics and the importance of fluctuations. It is well written.

---

## Editorial Decision

published